



# Kinetics of the OH + NO₂ reaction: Effect of water vapour and new parameterisation for global modelling.

Damien Amedro[1], Matias Berasategui[1], Arne J. C. Bunkan[1] Andrea Pozzer[1], Jos Lelieveld[1] and John N. Crowley[1]

[1]Division of Atmospheric Chemistry, Max-Planck-Institute for Chemistry, 55128 Mainz, Germany

*Correspondence to*: John N. Crowley (john.crowley@mpic.de)

**Abstract.** The effect of water vapour on the rate coefficient for the atmospherically important, termolecular reaction between OH and $NO_2$ was determined in $He$-$H_2O$ (277, 291 and 332 K) and $N_2$-$H_2O$ bath gases (292 K). Combining pulsed laser photolytic generation of OH and its detection by laser induced fluorescence (PLP-LIF) with in-situ, optical measurement of both $NO_2$ and $H_2O$ we were able to show that (in contrast to previous investigations) the presence of $H_2O$ increases the rate coefficient significantly. We derive a rate coefficient for $H_2O$ bath gas at the low-pressure limit ($k_0^{H2O}$) of $15.9 \times 10^{-30}$ $cm^6$ molecue$^{-2}$ s$^{-1}$. This indicates that $H_2O$ is a more efficient collisional quencher (by a factor of $\approx 6$) of the initially formed $HO$-$NO_2$ association complex than $N_2$ and a factor $\approx 8$ more efficient than $O_2$. Ignoring the effect of water-vapour will lead to an underestimation of the rate coefficient by up to 15% e.g. in the tropical boundary layer. Combining the new experimental results from this study with those from the companion paper in which we report rate coefficients obtained in $N_2$ and $O_2$ bath gases (Amedro et al., 2019) we derive a new parameterisation for atmospheric modelling of the OH + $NO_2$ reaction and use this in a chemical transport model (EMAC) to examine the impact of the new data on the global distribution of $NO_2$, $HNO_3$ and OH. Use of the new parameters (rather than those given in the IUPAC and NASA evaluations) result in significant changes in the $HNO_3$ / $NO_2$ ratio and NOx concentrations, the sign of which depends on which evaluation is used as reference. The model predicts the presence of HOONO (formed along with $HNO_3$ in the title reaction) in concentrations similar to those of $HO_2NO_2$ at the tropical tropopause.

## 1 Introduction

In our recent work on the title reaction (Amedro et al., 2019), we reported extensive measurements of the rate constant ($k_1$) for the termolecular reaction between OH and $NO_2$ (R1) in $N_2$ and $O_2$ bath gas over a large range of temperature and pressures.

$$OH + NO_2 + M \rightarrow HNO_3 + M \qquad \text{(R1a)}$$
$$\rightarrow HOONO + M \qquad \text{(R1b)}$$

Reaction (R1) converts $NO_2$ to nitric acid ($HNO_3$) and peroxynitrous acid (HOONO), and its rate strongly influences the relative abundance of atmospheric $NO_x$ ($NO_2 + NO$) and longer-lived "reservoirs" of $NO_x$ which include e.g. $HNO_3$ and organic nitrates. It also converts OH (the main initiator of atmospheric oxidation) to a long-lived reservoir, $HNO_3$. As the abundance of OH and $NO_x$ directly impact on photochemical ozone formation and the lifetimes of greenhouse gases, reaction (R1) may be considered one of the most important gas-phase processes in atmospheric science (Newsome and Evans, 2017). As outlined by Amedro et al. (2019), the rate coefficients and product-branching for this reaction are dependent on pressure and temperature and also on the bath-gas identity, i.e. the identity of the collision partner, M in reaction (R1). The per collision efficiency of energy transfer from the initially "hot" association complex to bath gas can vary considerably, with more complex bath gases molecules possessing more degrees of freedom and bonds with similar vibrational frequencies to those in the association complex being generally more efficient. In this sense, we may expect $H_2O$ to be better than $N_2$ or $O_2$ in quenching $[HO-NO_2]^\#$.





In this second part of our study of the reaction between OH and $NO_2$, we extend the experiments to $H_2O$ and He bath-gases. After $N_2$ ($\approx$ 78%) and $O_2$ ($\approx$ 21%) water vapour is the third most abundant gaseous species in the lower atmosphere. Its

concentration is highly variable in time and space, varying in mixing ratio from a few percent at sea level to parts-per-million in the stratosphere. Most of the atmosphere's water vapour is present in the planetary boundary layer where its average mixing ratio on the global scale is $\approx$ 1% but which may exceed 5% in tropical regions.

The effect of water vapour on gas-phase radical-reactions has been the subject of numerous studies (Buszek et al., 2011) and is sometimes interpreted in terms of formation of $H_2O$-radical complexes leading, via a chaperone type mechanism, to an

increase in the rate constant. An important example of this being the $HO_2$ self-reaction whereby the rate constant increases by a factor of up to two in the presence of water vapour due to formation of an $HO_2$-$H_2O$ complex (Lii et al., 1981; Kircher and Sander, 1984). On the other hand, the role of $H_2O$ as a collision partner in termolecular, atmospheric reactions has rarely been reported though its potential impact has been highlighted (Troe, 2003). Indeed, water vapour is known to be a more efficient third-body collider, by up to an order of magnitude compared to $N_2$ in termolecular reactions such as H + H + M, H + OH +

M and H + $O_2$ + M (Getzinger and Blair, 1969; Michael et al., 2002; Fernandes et al., 2008; Shao et al., 2019). Theoretical calculations (Allodi et al., 2006; Sadanaga et al., 2006; Thomsen et al., 2012) suggest that, under our experimental conditions, the fraction of OH and $NO_2$ clustered with $H_2O$ is < 0.1 % and they are unlikely to significantly impact on $k_1$.

The conclusions of three previous experiments examining the role of $H_2O$ in kinetic studies of reaction (R1) are highly divergent, with the addition of $H_2O$ found to 1) increase the rate coefficient (Simonaitis and Heicklen, 1972), 2) have no

measureable effect (D'Ottone et al., 2001) or 3) even reduce it (Sadanaga et al., 2006). The overall aim of this research was to clarify these differences and provide quantitative data on the third-body efficiency of $H_2O$ for the title reaction. Based on the kinetic data for the water vapour effect reported in this manuscript and in $N_2$ and $O_2$ presented in the first part of this study (Amedro et al., 2019) we have generated a new parameterisation for the overall rate coefficient, $k_1$, and examined its impact on atmospheric OH, $NO_X$ and $NO_Y$ in a global chemical transport model.

## 60   2. Experimental details

The details of the experimental set-up have been published previously (Wollenhaupt et al., 2000; Amedro et al., 2019) and only a brief description is given here.

### 2.1 PLP-LIF technique

The experiments were carried out in a quartz reactor of volume 500 $cm^{-3}$ which was thermostatted to the desired temperature

by circulating a 60:40 mixture of ethylene glycol-water. The pressure in the reactor was monitored with 100 and 1000 Torr capacitance manometers. Flow rates were chosen so that a fresh gas sample was available for photolysis at each laser pulse (laser frequency, 10 Hz), thus prevented build-up of products. Pulses of 248 nm laser light ($\approx$ 20 ns) for OH generation from $HNO_3$, $H_2O_2$ and $O_3$-$H_2O$ precursors were provided by an excimer laser (Compex 205 F, Coherent) operated using KrF.

| | | | |
|---|---|---|---|
| $HNO_3 + h\nu$ (248 nm) | $\rightarrow$ | $OH + NO_2$ | (R2) |
| 70   $H_2O_2 + h\nu$ (248 nm) | $\rightarrow$ | 2 OH | (R3) |
| $O_3 + h\nu$ (248 nm) | $\rightarrow$ | $O(^1D) + O_2$ | (R4) |
| $O(^1D) + H_2O$ | $\rightarrow$ | 2 OH | (R5) |

OH concentrations ($10^{11}$ – $10^{12}$ molecule $cm^{-3}$) were similar to those reported by Amedro et al. (2019) and the same arguments, which rule out significant influence of secondary reactions, apply. The concentration ranges of the $H_2O_2$, $HNO_3$ and $O_3$

precursors are listed in Tables 1 and 2.



OH was detected following excitation of the OH $A^2\Sigma(v'=1) \leftarrow X^2\Pi(v''= 0)$ transition (Q11(1) at 281.997 nm using a YAG-pumped dye laser (Quantel-Brilliant B and Lambda-Physik Scanmate). OH fluorescence was detected by a photomultiplier tube (PMT) screened by a 309 nm interference filter and a BG 26 glass cut-off filter.

### 2.2 On-line absorption measurement of $NO_2$ and $H_2O$ concentration

As discussed by Amedro et al. (2019),  the determination of the $NO_2$ concentration is critical for accurate measurement of $k_1$. We therefore deployed in-situ, broad-band (405 – 440 nm) and single wavelength (365 nm) optical absorption spectroscopy. The former was located prior (in flow) to the quartz-reactor, the latter was located behind the quartz-reactor. Using the broadband cell, the $NO_2$ concentration was retrieved by least square fitting from 405 to 440 nm to a reference spectrum (Vandaele et al., 2002) degraded to the resolution of our spectrometer. Simultaneously, we measured $NO_2$ at 365 nm using the

absorption cross-section $5.89 \times 10^{-19}$ $cm^2$ molecule$^{-1}$ determined previously by Amedro et al. (2019) who give a detailed description of the $NO_2$ concentration measurements and the choice of reference spectrum. For the temperatures used in this study, corrections to the $NO_2$ concentration due to formation of the $N_2O_4$ dimer were not necessary.

For the present experiments, a third absorption cell ($l = 40$ cm) was placed downstream of the quartz-reactor to measure the $H_2O$ concentration at 184.95 nm. This set-up used a low-pressure Hg-Penray lamp isolated with a 185 nm interference filter

as light source. Optical extinction was converted to concentrations using a cross-section of $7.14 \times 10^{-20}$ $cm^2$ molecule$^{-1}$ (Cantrell et al., 1997).

### 2.3 Chemicals

$N_2$ and He (Westfalen 99.999%) were used without further purification. $H_2O_2$ (AppliChem, 50 wt. %) was concentrated to >90 wt.% by vacuum distillation. Anhydrous nitric acid was prepared by mixing $KNO_3$ (Sigma Aldrich, 99%) and $H_2SO_4$ (Roth,

98%), and condensing $HNO_3$ vapour into a liquid nitrogen trap. NO (3.5 AirLiquide) was purified of other nitrogen oxides by fractional, vacuum distillation and then converted to $NO_2$ via reaction with a large excess of $O_2$. The $NO_2$ thus made was trapped in liquid $N_2$ and the excess $O_2$ was pumped out. The resulting $NO_2$ was stored as a mixture of ~0.5% $NO_2$ in $N_2$ or ~5.5% $NO_2$ in He. Distilled $H_2O$ (Merck, Liquid Chromatography grade) was degassed before use and kept at constant temperature.

## 3 Results and Discussion

### 3.1 Measurements of $k_1$ in He bath-gas and comparison with literature

The relatively low vapour pressure of water at typical atmospheric temperatures precludes working with pure $H_2O$ as bath gas. Our study of the role of $H_2O$ as collision partner in reaction (R1) was therefore carried out in mixtures of He-$H_2O$ and $N_2$-$H_2O$. In order to separate the effects of $H_2O$ and He, we also required accurate rate coefficients for pure He bath gas, which we

describe below. As for the $N_2$ and $O_2$ bath-gas datasets (Amedro et al., 2019), the experiments were carried out under pseudo-first-order conditions ($[NO_2] >> [OH]$) so that Eqn. 1-2 describe the decay of OH and the derivation of the bimolecular rate coefficient, $k_1$.

$$[OH]_t = [OH]_0 \exp(-k't) \tag{1}$$

where $[OH]_t$ is the concentration (molecule $cm^{-3}$) at time $t$ after the laser pulse. $k'$ is the pseudo-first order rate coefficient and

is defined as

$$k' = k_1[NO_2] + k_d \tag{2}$$

where $k_d$ ($s^{-1}$) accounts for OH-loss due to diffusion out of the reaction zone and reaction with its photolytic precursors such as $HNO_3$ or $H_2O_2$.



An exemplary dataset illustrating OH decays and a plot of $k$' versus [NO$_2$] is given in Fig. S1 of the supplementary
information).

Values of $k_1$ obtained in He bath-gas (25-690 Torr, 292 K) are summarised in Fig. 1 and 2 and listed in Table 1. The kinetics
of termolecular reactions can be described by the Lindemann-Hinshelwood mechanism whereby the rate constant at the low-
pressure limit ($k_0$, units in cm$^6$ molecule$^{-2}$ s$^{-1}$) is proportional to the pressure and at the high pressure limit ($k_\infty$, units in cm$^3$
molecule$^{-1}$ s$^{-1}$) is independent of pressure. In the intermediate pressure range, the fall-off regime, the rate coefficient is a
function of both low-pressure ($k_0$) and high-pressure ($k_\infty$) rate coefficients and the (reaction-partner dependent) broadening
factor $F$ which accounts for the lower rate constant measured in the fall-off regime than predicted by the Lindemann-
Hinshelwood mechanism reactions (Troe, 1983).

$$k = \frac{k_0[\text{M}]k_\infty}{k_0[\text{M}] + k_\infty} F \tag{3}$$


The solid lines in Figs. 1 and 2 are fits according to the Troe formalism for termolecular reactions (Troe, 1983) as adopted by
the IUPAC panel in their evaluation atmospheric reactions:

$$k(P,T) = \frac{k_0\left(\frac{T}{300}\right)^{-m}[\text{M}]k_\infty\left(\frac{T}{300}\right)^{-n}}{k_0\left(\frac{T}{300}\right)^{-m}[\text{M}] + k_\infty\left(\frac{T}{300}\right)^{-n}} F \tag{4}$$

where $T$ is the temperature in Kelvin, [M] is the bath-gas concentration in molecule cm$^{-3}$, $m$ and $n$ are dimensionless
temperature exponents.

The broadening factor, $F$, is:

$$\log F = \frac{\log F_c}{1 + \left[\log\left(\frac{k_0\left(\frac{T}{300}\right)^{-m}[\text{M}]}{k_\infty\left(\frac{T}{300}\right)^{-n}}\right)/N\right]^2} \tag{5}$$

Where $N = [0.75 - 1.27 \log F_c]$ and $F_c$ is the broadening factor at the centre of the fall-off curve.
As discussed in some detail in the first part of our studies of the title reaction (Amedro et al., 2019), the low- or high-pressure
rate constants for the title reaction ($k_0$ and $k_\infty$) are not well defined by existing data sets, which do not deliver sufficiently
accurate rate coefficient at very low pressures (< 1 mbar) or at very high pressures (> 500 bar). Studies in which $k_\infty$ has been
derived from rates of vibrational relaxation of OH (Smith and Williams, 1985; D'Ottone et al., 2005), return values of $k_\infty$ that
provide some constraint on its value, but the associated uncertainty is too large to consider this parameter well defined.

In our first paper, Amedro et al. (2019) describe highly accurate measurements of $k_1$ over a wide range temperatures and
pressures in the fall-off regime. From measurements of $k_1$ in N$_2$ bath-gas, we retrieved values of $k_0$ and $k_\infty$ of $2.6 \times 10^{-30}$ cm$^6$
molecule$^{-2}$ s$^{-1}$ and $6.3 \times 10^{-11}$ cm$^3$ molecule$^{-1}$ s$^{-1}$, respectively, by fixing $F_c$ to a value of 0.39 which has theoretical basis (Cobos
and Troe, 2003). The reasons for choosing this value of $F_c$ are discussed in Amedro et al. (2019). Note that whereas $k_0$ is
dependent on the bath-gas used, at the high-pressure limit, $k_\infty$ should be the same in N$_2$, O$_2$, He or H$_2$O bath gases.

In Fig.1 we display pressure dependent rate coefficients (solid, black squares) obtained in He bath-gas at 292 K. The black
line is a fit (Eqn. 4) to our data with $k_\infty$ fixed to $6.3 \times 10^{-11}$ cm$^3$ molecule$^{-1}$ s$^{-1}$ and $n = 0$ as derived from an extensive dataset
obtained using N$_2$ bath-gas (Amedro et al., 2019). For this dataset, the best fit is obtained with $F_c = 0.32$, and $k_0^{\text{He}} = 1.4 \times 10^{-30}$
cm$^6$ molecule$^{-2}$ s$^{-1}$. When using $F_c = 0.39$ (i.e. same value as that obtained in N$_2$ bath-gas) the fit slightly overestimates (~5 %)
the measurements at pressures above ~300 Torr whereas it underestimates by 10 % at lower pressures (Fig. S2). The high
precision of our measurements in He and N$_2$ indicates that different broadening factors ($F_c$) are required to interpret the pressure
dependence of $k_1$ obtained in N$_2$ and He. This can be rationalized by considering that $F_c$ is the product of strong-collision ($F_c^{SC}$)
and weak-collision ($F_c^{WC}$) components (Eqn. 6-8) (Gilbert et al., 1983; Troe, 1983; Troe and Ushakov, 2011)

$$F_c \approx F_c^{SC} F_c^{WC} \tag{6}$$





$$F_c^{SC} \approx S_K^{-0.62} \approx \left(1 + \frac{r}{2}\right)^{-0.62} \tag{7}$$

$$F_c^{WC} \approx \beta_c^{0.14} \tag{8}$$

Here, $S_K$ is the Kassel parameter and $r$ is the total number of external rotational modes of the reactants (equal to 5 in the reaction between OH and $NO_2$) and $\beta_c$ is the collision efficiency. While the strong collision component is independent of bath gas ($F_c^{SC} \approx 0.46$ for the title reaction) a change in $F_c^{WC}$ due to a lower collision efficiency ($\beta_c$) of He relative to $N_2$ is likely. The collision efficiency for $N_2$ which was used to calculate $F_c = 0.39$ was $\beta_c(N_2) \approx 0.3$ (Troe, 2001). The value of $F_c = 0.32$

from our He data implies $\beta_c(\text{He}) \approx 0.08$, a factor 3.7 times lower than $\beta_c(N_2)$. A large difference in collision efficiency between $N_2$ and He is consistent with theoretical calculations (Glänzer and Troe, 1974; Troe, 2001; Golden et al., 2003).

In Fig 1, we also compare our measurements of $k_1$ in He with data collected in the same pressure range using similar techniques. The three first measurements (Morley and Smith, 1972; Anastasi and Smith, 1976; Wine et al., 1979) used flash photolysis of $H_2O$ as a OH precursor with detection of OH by resonance fluorescence. Morley and Smith (1972) reported rate coefficients

at pressures of 20 to 280 Torr at room temperature with the $NO_2$ concentration calculated manometrically. Our parametrisation agrees within the combined uncertainty of both measurements (Figure S3). Anastasi and Smith (1976) reported one value of $k_1$ at 25 Torr He which is ≈ 20 % lower than our measurement. Wine et al. (1979) presented values of $k_1$ at 3 pressures of He. The agreement with our parameterisation at the lowest two pressures is excellent but a deviation of ≈ 20% is observed at the highest pressure (Figure S4). As both studies measured $NO_2$ concentrations using optical absorption at 365 nm, the ≈ 20%

difference is significant. Most recently, D'Ottone et al. (2001) reported rate coefficients from 30 to 600 Torr He using a very similar approach to ours i.e. PLP-LIF technique with in situ measurements of $NO_2$ by absorption at 365 nm. The disagreement (up to 40%) between our measurements and theirs exceed the combined reported uncertainty (Figure S5). While it is unclear what could have caused the discrepancy, we note that the data of D'Ottone et al. (2001) are significantly more scattered and do not describe a smooth increase in rate coefficient with pressure as expected from termolecular reactions in the fall-off

regime. This would appear to indicate an underestimation of the total uncertainty in their study.

Figure 2 extends the pressure range to additionally display data obtained in low pressure flow-tubes (Westenberg and Dehaas, 1972; Anderson et al., 1974; Erler et al., 1977; Anderson, 1980) and the high-pressure measurements by Hippler et al. (2006). At low pressures we are in excellent agreement (within 10%) with the data of Erler et al. (1977) but predict values ≈ 40% lower than those reported by Westenberg and Dehaas (1972) and Anderson (1980). The data of Anderson et al. (1974) display

a large intercept ($4.9 \times 10^{-14}$ cm$^3$ molecule$^{-1}$ s$^{-1}$) at zero pressure, which is attributed to a second-order heterogeneous removal rate constant. As indicated in a critical assessment of the low-pressure data by Amedro et al. (2019) it is unclear whether one can simply subtract a constant value equal to the intercept (obtained from a linear fit) to each data point. If we were to do so, the work by Anderson et al. (1974) would be in very good agreement with the low pressure study by Erler et al. (1977) as well as with our parametrisation extended to low pressures. Additionally, Amedro et al. (2019) demonstrated that, owing to the

large, asymmetric broadening of fall-off for this reaction the assumption that the rate coefficient is in the low pressure limit at $N_2$ pressures of 0.5 Torr $< p <$ 10 Torr is invalid and leads to underestimation of $k_0$. This observation is still true of datasets obtained at low pressures of He, so that while very good agreement is observed between our parametrisation and individual rate coefficients obtained between 3 and 8 Torr of He, reported values of $k_0^{\text{He}}$ are 40 % lower than our values obtained from the fall-off analysis. As indicated in Fig. 2, our parametrisation of $k_1$ in He is in very good agreement with the high pressure

data reported by Hippler et al. (2006).

### 3.2 Influence of $H_2O$ on $k_1$

As mentioned above, the effect of water vapour on $k_1$ was determined in mixtures of $H_2O$ with both $N_2$ and He. This is simply because the vapour pressure of $H_2O$ at room temperature (≈ 17 Torr at 293 K) is too low to enable experiments in pure $H_2O$ bath gas to be conducted using our instrument. The measurements were performed at low density ([M] = $1.6 \times 10^{18}$ molecule



cm$^{-3}$; 50 Torr at 293 K) where the relative increase of $k_1$ in the presence of $H_2O$ is pronounced, resulting in greater accuracy in the determination of $k_0^{H2O}$. Experimental data on the influence of $H_2O$ on $k_1$ was obtained in $N_2$-$H_2O$ and He-$H_2O$ mixtures by varying the $H_2O$ mixing ratio, $x_{H2O}$, from 0.05 to 0.27 ([$H_2O$] = 0.9 - $4.5 \times 10^{17}$ molecule cm$^{-3}$) while keeping the total pressure constant at 50 Torr. Under these conditions, the addition of $H_2O$ resulted in an increase in $k_1$ up to a factor of two as illustrated by the datasets of Fig. 3 in which the increase in slope as more water-vapour is added is proportional to the increase in $k_1$ (Eqn. 2). At the highest concentration of water vapour ($4.5 \times 10^{17}$ molecule cm$^{-3}$) the rate coefficient in He-$H_2O$ increased by > factor 3 compared to that obtained in pure He (see Table 1).

In order to determine the temperature dependence of the enhancement in $k_1$ caused by the presence of water, the experiments in He were carried out at 3 different temperatures (277, 291 and 332 K). The values of $k_1$ obtained from these experiments are plotted versus the mole-fraction of $H_2O$ in Fig. 4b. At the pressures used in our experiments, $k_1$ displays fall-off, precluding direct measurement of $k_0^{H2O}$.

In other words, the total rate constant measured in e.g. a $H_2O$-$N_2$ bath gas is not equal to the sum of the individual rate constants calculated from the single mixing ratios of $N_2$ and $H_2O$ i.e. $k_{N2\text{-}H2O} \neq k_{N2} + k_{H2O}$. $k_{N2\text{-}H2O}$ is only equal to the sum of $k_{N2}$ and $k_{H2O}$ at the low pressure limit (<< 1 Torr in the case of the OH reaction with $NO_2$) and under certain conditions where gas mixtures are composed of strong colliders molecules and/or have similar collision efficiencies (Troe, 1980; Burke and Song, 2017). Additionally, at the high-pressure end of the fall-off curve, the rate coefficient is independent of bath gas composition.

To be able to make a reasonable prediction of this effect under atmospheric conditions where the mole fraction of water vapour, $x_{H2O}$, can be as large as 0.05, we analysed our measurements using two different approaches to determine $k_0^{H2O}$. In the first case (simple mixing), the low pressure rate constant in a $N_2$-$H_2O$ mixture is defined as the sum of two individual low pressure limit rate constants,

$$k(P, T) = \frac{\left(x_{N2}k_0^{N2}\left(\frac{T}{300}\right)^{-m} + x_{H2O}k_0^{H2O}\left(\frac{T}{300}\right)^{-o}\right)[M]k_\infty\left(\frac{T}{300}\right)^{-n}}{\left(x_{N2}k_0^{N2}\left(\frac{T}{300}\right)^{-m} + x_{H2O}k_0^{H2O}\left(\frac{T}{300}\right)^{-o}\right)[M] + k_\infty\left(\frac{T}{300}\right)^{-n}} F \qquad (9)$$

where $x_{N2}$ and $x_{H2O}$ are the mixing ratio for $N_2$ and $H_2O$ respectively, $k_0^{N2}$ and $k_0^{H2O}$ are low-pressure limiting rate constants (units of cm$^6$ molecule$^{-2}$ s$^{-1}$) for pure $N_2$ and $H_2O$, $k_\infty$ is the high-pressure limit rate constant (units of cm$^3$ molecule$^{-1}$ s$^{-1}$), $T$ is the temperature in Kelvin, [M] is the molecular density (molecule cm$^{-3}$) and $m$, $n$ and $o$ are dimensionless temperature exponents.

The broadening factor, $F$, is:

$$\log F = \frac{\log F_c}{1 + \left[\log\left(\frac{\left(x_{N2}k_0^{N2}\left(\frac{T}{300}\right)^{-m} + x_{H2O}k_0^{H2O}\left(\frac{T}{300}\right)^{-o}\right)[M]}{k_\infty\left(\frac{T}{300}\right)^{-n}}\right)/N\right]^2} \qquad (10)$$

Where $N = [0.75 - 1.27 \log F_c]$ and $F_c$ is the broadening factor at the centre of the fall-off curve.

In the second approach (linear mixing), we follow Burke and Song (2017) where, additionally to the low pressure limiting rate coefficients, the broadening factors for each bath gas are also mixed linearly and $\log F^{N2\text{-}H2O}$ is defined as

$$\log F^{N2\text{-}H2O} = \tilde{X}_{N2} \log F^{N2} + \tilde{X}_{H2O} \log F^{H2O} \qquad (11)$$

where $\quad \tilde{X}_{N2} = \frac{x_{N2}k_0^{N2}\left(\frac{T}{300}\right)^{-m}[M]}{\left(x_{N2}k_0^{N2}\left(\frac{T}{300}\right)^{-m} + x_{H2O}k_0^{H2O}\left(\frac{T}{300}\right)^{-o}\right)[M]}; \quad \tilde{X}_{H2O} = \frac{x_{H2O}k_0^{H2O}\left(\frac{T}{300}\right)^{-m}[M]}{\left(x_{N2}k_0^{N2}\left(\frac{T}{300}\right)^{-m} + x_{H2O}k_0^{H2O}\left(\frac{T}{300}\right)^{-o}\right)[M]} \qquad (12)$

$$\log F^{N2} = \frac{\log F_c^{N2}}{1 + \left[\log\left(\frac{\left(x_{N2}k_0^{N2}\left(\frac{T}{300}\right)^{-m} + x_{H2O}k_0^{H2O}\left(\frac{T}{300}\right)^{-o}\right)[M]}{k_\infty\left(\frac{T}{300}\right)^{-n}}\right)/\left(0.75 - 1.27 \log F_c^{N2}\right)\right]^2} \qquad (13)$$





$$\log F^{H2O} = \frac{logF_c^{H2O}}{1+\left[log\left(\frac{\left(x_{N2}k_0^{N2}\left(\frac{T}{300}\right)^{-m}+x_{H2O}k_0^{H2O}\left(\frac{T}{300}\right)^{-o}\right)[M]}{k_\infty\left(\frac{T}{300}\right)^{-n}}\right)/(0.75-1.27\log F_c^{H2O})\right]^2} \qquad (14)$$

where $F_c^{N2}$ and $F_c^{H2O}$ are the broadening factor at the centre of the fall off curve for $N_2$ and $H_2O$.

In the case where two bath gases have identical (or very similar) values of $F_c$, the two approaches result in identical predictions

and the first approach will be preferred for its simplicity. This is the case for $N_2$ and $H_2O$ bath gases. However, when two bath-gases have significantly different values of $F_c$ (as is the case for He-$H_2O$ mixtures, see below) the second approach provides a more accurate parameterisation.

### 3.2.1 Parameterisation of $k_1$ from data obtained in $N_2$-$H_2O$ and He-$H_2O$ bath gases

Values of $k_1$ obtained in $N_2$–$H_2O$ and He-$H_2O$ bath gases are listed in Table 2. Each rate coefficient obtained in $N_2$-$H_2O$ bath

gas was defined by 5 parameters: the mixing ratio of $N_2$ and $H_2O$ ($x_{N2}$ and $x_{H2O}$) the overall rate coefficient ($k_1$) the molecular density [M] and the temperature $T$. We performed a multivariate fit of the $N_2$-$H_2O$ dataset with $k_0^{H2O}$ as variable, all other parameters fixed with: $k_\infty = 6.3 \times 10^{-11}$ cm$^3$ molecule$^{-1}$ s$^{-1}$, $k_0^{N2} = 2.6 \times 10^{-30}$ cm$^6$ molecule$^{-2}$ s$^{-1}$ and $m = 3.6$ as derived in Amedro et al. (2019), $o$ was fixed to 3.4 (see below) and $F_c$ was held at 0.39 making the assumption that the broadening factors at the centre of the fall-off curve for $H_2O$ and $N_2$ were identical (simple mixing scenario, see above). The fit to the data returned

$k_0^{H2O} = (15.9 \pm 0.7) \times 10^{-30}$ cm$^6$ molecule$^{-2}$ s$^{-1}$ where the uncertainty is $2\sigma$ (statistical only). The solid black line on the upper panel of Fig. 4a represents the parametrisation for a varying fraction of $H_2O$ in $N_2$ at a total pressure of 50 Torr using the parameters given above. Equating $F_c^{H2O}$ and $F_c^{N2}$ simplifies the analysis, though it is likely that $F_c^{H2O} > F_c^{N2}$ as the collision efficiency ($\beta_c$) is likely to be larger for $H_2O$ than for $N_2$.

We found that the He-$H_2O$ data cannot be modelled using the "single mixing" approach assuming the same $F_c$ for both He and

$H_2O$ bath gas and the linear mixing approach was therefore preferred. In order to analyse the data we fixed the following parameters: $k_0^{H2O} = 15.9 \times 10^{-30}$ cm$^6$ molecule$^{-2}$ s$^{-1}$, $F_c^{H2O} = 0.39$, $F_c^{He} = 0.32$ and $k_0^{He} = 1.4 \times 10^{-30}$ cm$^6$ molecule$^{-2}$ s$^{-1}$ and $m = 3.1$ to derive $o = (3.4 \pm 0.8)$ ($2\sigma$, statistical only), which describes the temperature dependent of the low pressure limit in $H_2O$ as depicted in Fig. 4b.

There is clearly some uncertainty related to the arbitrary use of $F_c^{H2O} = 0.39$. For example, if we were to use analyse the data

in $N_2$-$H_2O$ using $F_c^{H2O} = 0.6$ and the linear mixing method we retrieve $k_0^{H2O} = 10 \times 10^{-30}$ cm$^6$ molecule$^{-2}$ s$^{-1}$, which is $\approx 50\%$ lower than our preferred value. The effect of the different analyses can be assessed by comparing the predicted impact of $H_2O$ on $k_1$ at 80% relative humidity, 1000 mbar and 313 K. If we set $F_c^{H2O} = 0.39$ we predict that the effect of $H_2O$ is to increase $k_1$ by 15% while choosing $F_c^{H2O} = 0.6$ results in an increase of 20%. Theoretical calculation of the relative values of Fc in $N_2$, $O_2$ and $H_2O$ bath gases input would be useful to reduce this uncertainty. Our data indicate a significant, positive trend in $k_1$ when

adding $H_2O$. As discussed above, more efficient energy transfer from [HO-NO$_2$]$^\#$ in collision with $H_2O$ compared to $N_2$ is intuitive and supported by the present dataset as well as that of Simonaitis and Heicklen (1972) who derived $k_0^{H2O} = 11 \times 10^{-30}$ cm$^6$ molecule$^{-2}$ s$^{-1}$. Given the complexity of the analysis, this may be considered to be in good agreement. This result is however not consistent with the observations of D'Ottone et al. (2001) who report no significant change in $k_1$ in 150 Torr of He when adding either 10 or 20 Torr of $H_2O$ and is completely at odds with the conclusions of Sadanaga et al. (2006), who report a

reduction in $k_1$ (by 18%) when adding 29.1 mbar of $H_2O$ at atmospheric pressure. If our value for $k_0^{H2O}$ is correct, D'Ottone et al. (2001) should have seen an increase in $k_1$ of $\approx 55\%$ and Sadanaga et al. (2006) should have observed an increase of $\approx 5\%$. A potential explanation for the very divergent observations of the effect of $H_2O$ is the heterogeneous loss of $NO_2$ when adding $H_2O$. We tested for $NO_2$ loss in a set of experiments in which $NO_2$ and $H_2O$ were monitored simultaneously while systematically varying the amount of $H_2O$. Our results indicated a reduction in the concentration of $NO_2$ by up to $\approx 20\%$ as we

increased the concentration of $H_2O$ up to $4.5 \times 10^{17}$ molecule cm$^{-3}$. Unless $NO_2$ is monitored in-situ (as in our experiments),





20% loss of $NO_2$ would lead to a similar size reduction in the OH decay constant and thus an underestimation of the rate coefficient. A fractional loss of $NO_2$ of this magnitude would explain why Sadanaga et al. (2006) found an apparent reduction in $k_1$ when adding $H_2O$.

However, the situation becomes more complex if $NO_2$ is converted to trace gases that are reactive towards OH. For this reason,

we performed an additional experiment to investigate whether $NO_2$ was converted via reaction with $H_2O$ on surfaces to HONO and/or $HNO_3$. Note that conversion of $NO_2$ to HONO at low pressures (e.g. 50 Torr) would result in an increase in the OH decay constant ($k_{OH+HONO} > k_{OH+NO2}$), whereas conversion of $NO_2$ to $HNO_3$ would result in a decrease ($k_{OH+HNO3} < k_{OH+NO2}$). In order to test for the presence of HONO, we modified the broadband absorption set-up by replacing the halogen lamp with a deuterium lamp, allowing us to detect HONO around 350 nm as well as $NO_2$. The optical absorption of $NO_2$ and HONO

(340 – 380 nm) was monitored in a flow of $NO_2$ ($1.7 \times 10^{15}$ cm$^{-3}$) at 50 Torr He in the absence and presence of $H_2O$ ([$H_2O$] = $4.5 \times 10^{17}$ molecule cm$^{-3}$, the maximum concentration used in this work). A depletion in $NO_2$ of 21% ($3.7 \times 10^{14}$ molecule cm$^{-3}$) was observed when $H_2O$ was added. An analysis of the spectra with and without $H_2O$ (Fig. S6) enabled us to establish an upper limit to the HONO concentration of $\approx 1 \times 10^{13}$ molecule cm$^{-3}$, which would corresponds to just 3% of the $NO_2$ lost. At this concentration, HONO does not significantly increase the loss-rate of OH (< 3% using a rate coefficient for reaction of OH

with HONO of $6.0 \times 10^{-12}$ cm$^3$ molecule$^{-1}$ s$^{-1}$ (IUPAC, 2019)). In the same experiment, we also recorded the optical density at 185 nm where $H_2O$, $NO_2$ and $HNO_3$ all absorb. Despite the large $HNO_3$ absorption cross-section at this wavelength ($1.6 \times 10^{-17}$ cm$^2$ molecule$^{-1}$, Dulitz et al. (2018)) we found no evidence for $HNO_3$ formation, indicating that the $NO_2$ lost was not converted to gas-phase $HNO_3$. Given its great affinity for glass in the presence of $H_2O$, we expect that any $HNO_3$ formed is strongly partitioned to the walls of the reactor. The tests indicate that, on the time scales of our experiments, $NO_2$ is lost

irreversibly on the humidified walls of our experiment. The maximum concentration of $H_2O$ used in this experiment, $4.5 \times 10^{17}$ molecule cm$^{-3}$, corresponding to a relative humidity of 80% (at 292 K) so that $H_2O$ condensation is not expected.

It is difficult to establish whether our observations of significant $NO_2$ loss can explain the result of D'Ottone et al. (2001), who did not observe an enhancement in $k_1$. D'Ottone et al. (2001) did not state whether, in their experiments, $NO_2$ and $H_2O$ were monitored simultaneously. Also, our observed loss of $NO_2$ is not necessarily transferable to other studies as the heterogeneous

loss of $NO_2$ will vary from one experimental set-up to the next, as residence times and surface areas may vary substantially.

A very simple calculation serves to illustrate the role of water vapour as a third-body quencher for the title reaction. We consider e.g. the tropical boundary layer with a temperatures of 30 °C and a relative humidity of 80% at a total pressure of 1 bar. The pressure of water vapour is 34 mbar, those of $O_2$ and $N_2$ are then 203 and 753 mbar, respectively. A rough contribution of each quenching gas to the overall rate coefficient can be calculated from the respective low-pressure rate coefficients. For

$N_2$, $O_2$ and $H_2O$ these are (in units of $10^{-30}$ cm$^3$ molecule$^{-1}$ s$^{-1}$) 2.6, 2.0 and 15.9. Water vapour is therefore a factor $\approx 8$ more efficient than $O_2$. For our tropical boundary layer case-study, in which the $O_2$ pressure is only a factor of six greater than that of $H_2O$, we calculate that $H_2O$ contributes more to the rate coefficient of the title reaction than does $O_2$. Clearly, the neglect of including the quenching effect of $H_2O$ leads to underestimation of the rate coefficient for this centrally important atmospheric reaction.

In order to assess both the effect of $H_2O$ (this work) and the new parameterisation for $k_1$ in $N_2$ and $O_2$ bath-gases presented in first part of this study (Amedro et al., 2019), we have used a 3D chemical transport model (EMAC, see below) to explore the impact on a global scale.

### 3.3. Atmospheric modelling of the OH + NO₂ reaction including the effect of water vapour

The EMAC (ECHAM-MESSy Atmospheric Chemistry) model employed is a numerical chemistry and climate simulation

system (Jöckel et al., 2006; Jöckel et al., 2010) using the 5th generation European Centre Hamburg general circulation model (ECHAM5, Roeckner et al. (2006)) as core atmospheric general circulation model. For the present study we applied EMAC





(ECHAM5 version 5.3.02, MESSy version 2.53.0) in the T42L47MA-resolution, i.e. with a spherical truncation of T42 (corresponding to a quadratic Gaussian grid of approx. 2.8 by 2.8 degrees in latitude and longitude) with 47 vertical hybrid pressure levels up to 0.01 hPa. The model has been weakly nudged in spectral space, nudging temperature, vorticity, divergence

and surface pressure (Jeuken et al., 1996). The chemical mechanism scheme adopted (MOM, Mainz Organic Mechanism) includes oxidation of isoprene, saturated and unsaturated hydrocarbons, including terpenes and aromatics (Sander et al., 2019). Further, tracer emissions and model set-up are similar to the one presented in Lelieveld et al. (2016a). EMAC model predictions have been evaluated against observations on several occasions (Pozzer et al., 2010; de Meij et al., 2012; Elshorbany et al., 2014; Yoon and Pozzer, 2014): For additional references, see http://www.messy-interface.org. For this study, EMAC was used

in a chemical-transport model (CTM mode) (Deckert et al., 2011), i.e., by disabling feedbacks from photochemistry on radiation and dynamics. Two years were simulated (2009-2010), with the first year used as spin-up time.

The following parameterisation of $k_1$ was implemented in EMAC; values of each parameter are listed in Table 3.

$$k_1(P,T) = \frac{\left(x_{N2}k_0^{N2}\left(\frac{T}{300}\right)^{-m} + x_{O2}k_0^{O2}\left(\frac{T}{300}\right)^{-q} + x_{H2O}k_0^{H2O}\left(\frac{T}{300}\right)^{-o}\right)Mk_\infty\left(\frac{T}{300}\right)^{-n}}{\left(x_{N2}k_0^{N2}\left(\frac{T}{300}\right)^{-m} + x_{O2}k_0^{O2}\left(\frac{T}{300}\right)^{-q} + x_{H2O}k_0^{H2O}\left(\frac{T}{300}\right)^{-o}\right)M + k_\infty\left(\frac{T}{300}\right)^{-n}}F \quad (15)$$

The broadening factor, $\log F$, is:

$$\log F = \frac{\log F_c}{1 + \left[\log\left(\frac{\left(x_{N2}k_0^{N2}\left(\frac{T}{300}\right)^{-m} + x_{O2}k_0^{O2}\left(\frac{T}{300}\right)^{-q} + x_{H2O}k_0^{H2O}\left(\frac{T}{300}\right)^{-o}\right)M}{k_\infty\left(\frac{T}{300}\right)^{-n}}\right)/[0.75 - 1.27\log F_c]\right]^2} \quad (16)$$

As described in Section 1, the reaction between OH and $NO_2$ forms not only $HNO_3$ but also HOONO. HOONO decomposes rapidly at typical boundary layer temperatures but is long lived with respect to thermal dissociation at the temperatures found

in the upper troposphere and lower stratosphere (UTLS).

HOONO + M $\rightarrow$ OH + $NO_2$ (R6)

The rate constant ($k_6$) for thermal decomposition of HOONO was calculated from the channel specific rate coefficient for its formation ($k_{1\alpha}$) and an equilibrium coefficient: $k_6 = k_{1\alpha} / K_{eq}$, where $K_{eq} = 3.5 \times 10^{-27} \exp(10135/T)$ (Burkholder et al., 2015; IUPAC, 2019) based on the analysis of (Golden et al., 2003). The branching ratio to HOONO formation ($\alpha$) was adapted from

the present IUPAC recommendations for $k_{1a}$ and $k_{1b}$ which were derived from experimental work (Hippler et al., 2006; Mollner et al., 2010) and theoretical analysis (Troe, 2012). The IUPAC recommendations were augmented with a pressure independent HOONO yield of 0.035 to better represent the dataset of Mollner et al. (2010) who detected HOONO directly at room temperature. The expression used and a plot of $\alpha$ at different temperatures and pressures is given in Fig. S7 of the supplementary information.

In the absence of experimental data on the reactions of HOONO with OH or on its photolysis, we follow the approach of Golden and Smith (2000) and set these equal to those for $HO_2NO_2$ :

HOONO + OH $\rightarrow$ $HO_2$ + $NO_3$ (R7)

HOONO + $h\nu$ $\rightarrow$ HO + $NO_2$ (R8)

In Fig. 5, we illustrate the global impact (annual average) of $H_2O$-vapour on the rate coefficient. We plot the fractional

reduction in $k_1$ at the Earth's surface when setting $x_{H2O}$ to zero rather than using the EMAC global water-vapour fields. We focus on the boundary layer as the $H_2O$ concentration is largest here and decreases rapidly with altitude.

As expected, the greatest effect is found in warm, tropical regions where neglecting the impact of water vapour results in an average underestimation of the rate coefficient by up to $\approx 8$ %. At higher/lower latitudes the effect is diminished and water vapour accounts for only 3-4 % of the overall rate coefficient at 40 ° N/S.

Our experimental data do not give insight into whether the $H_2O$-induced enhancement in $k_1$ is accompanied by a change in the branching ratio to favour either $HNO_3$ or HOONO. However, as the formation of HOONO is favoured at high pressures (more



effective collisional deactivation) it is possible that the HOONO yield may be enhanced relative to $HNO_3$ in the presence of $H_2O$. If this is the case, the increase in rate coefficient at high water vapour levels (e.g. in the tropical lower troposphere) may be to some extent offset by the subsequent thermal dissociation of HOONO in these warm regions.

As described by (Amedro et al., 2019) (Fig. 1 of their manuscript) two expert panels (IUPAC, NASA) evaluating kinetic data for use in atmospheric modelling fail to reach consensus for the title reaction, with the preferred rate coefficients differing by as much as 50% in the cold UTLS. For this reason we have calculated values of $\frac{k_1^{NASA}}{k_1^{this\ work}}$ and $\frac{k_1^{IUPAC}}{k_1^{this\ work}}$ at different altitudes and latitudes (i.e. at different temperatures and pressures). We parameterized the rate coefficient using the expressions given in this work (Eqn. 15, Table 3) and in the latest evaluations of IUPAC ($k_1$ last evaluated in 2017 (IUPAC, 2019)) and NASA

(last evaluation published in 2015 (Burkholder et al., 2015)). As displayed in Fig. 6, values of $\frac{k_1^{NASA}}{k_1^{this\ work}}$ and $\frac{k_1^{IUPAC}}{k_1^{this\ work}}$ vary greatly with pressure and temperature and thus altitude. The NASA recommendations are always slightly lower but in good agreement ($\leq 10\%$) for most of the troposphere, with larger differences (($\frac{k_1^{NASA}}{k_1^{this\ work}}$) always < 1) only observed in the lower and mid-stratosphere. At altitudes above $\approx 30$ km the ratio decreases to $\approx 0.8$. A comparison with the rate coefficient derived from the IUPAC parameterization, shows that $\frac{k_1^{IUPAC}}{k_1^{this\ work}}$ varies from $\approx 0.9$ at the surface to $\approx 1.1$ at the tropopause but increases

to >1.3 at the low pressures and temperatures that reign at 30 km and above. At high altitudes (low pressure and temperature) the rate coefficients that the evaluation panels recommend are strongly biased by choice of the rate coefficient (and its temperature dependence) at the low pressure limit. As discussed by Amedro et al. (2019) the available experimental data at low pressures and temperature are not of sufficient accuracy to use as basis for recommendation of $k_0$ and this is reflected in the highly divergent values of $k_1$ under these conditions.

As mentioned above, the atmospheric $HNO_3$ / $NO_2$ ratio is expected to be highly sensitive to the rate coefficient $k_1$, with an increase in $k_1$ resulting in an increase in the $HNO_3$ / $NO_2$ ratio and vice versa. The $HNO_3$ / $NO_2$ ratio also depends on the concentration of OH and thus the effect of using different values of $k_1$ will be most apparent in regions where the greatest OH concentrations are found, i.e. at low latitudes. At higher latitudes, especially in winter months where solar insolation is weak and OH levels are relatively low, the $HNO_3$ / $NO_2$ ratio will also be impacted by nighttime conversion of $NO_2$ to $N_2O_5$ and

finally, via heterogeneous hydrolysis, to $HNO_3$. In Fig. 7 we plot zonally and yearly averaged model values of $\frac{HNO3}{NO2}$ (IUPAC)/$\frac{HNO3}{NO2}$ (this work) in the upper panel and $\frac{HNO3}{NO2}$ (NASA)/$\frac{HNO3}{NO2}$ (this work) in the lower panel. Compared to the present parameterization of $k_1$, the IUPAC evaluation returns $HNO_3$ / $NO_2$ ratios that are between 0.9 and 1 throughout most of the lower and free-troposphere (up to $\approx 5$ km) and larger $HNO_3$ / $NO_2$ ratios (factor of 1.1 to 1.15) above $\approx 10$ km especially at the tropical tropopause. The divergence between the $HNO_3$ / $NO_2$ ratios increases as we move further into the stratosphere

with $\frac{HNO3}{NO2}$ (IUPAC)/$\frac{HNO3}{NO2}$ (this work) as large as 1.2 to 1.3 above 25 km. At the same time, $NO_X$ levels ($NO_X = NO + NO_2$) decrease by a factor $\approx 0.95$ (see Fig. S8 of the supplementary information). When we compare our parameterization with that of the NASA panel, the picture is largely reversed (lower panel). Again, we find reasonable agreement in the $HNO_3$ / $NO_2$ ratio in the lowermost atmosphere, but in this case lower values (0.8 to 0.9) in the lower stratosphere which are accompanied by a factor 1.06 change in $NO_X$ concentrations Fig. S8. For both the NASA and IUPAC parameterizations, the largest differences

in the $HNO_3$ / $NO_2$ ratio compared to the present study are found higher in the atmosphere. The modelling studies confirm the simple calculation of Amedro et al. (2019) (see Fig. 1 of their paper), showing that the IUPAC and NASA parameterizations result in very different values of $k_1$ in some parts of the atmosphere and will result in divergent predictions of partitioning of reactive nitrogen between $NO_X$ and $NO_Y$. Use of the parameterization based on the present dataset lies roughly between the two evaluations, with best agreement observed with NASA for the lower atmosphere. However, as previous laboratory studies

had not identified the important role of $H_2O$ in the title reaction, which could therefore not be incorporated in either of the



previous parameterizations, any agreement at better than 10% level is fortuitous, reflecting random cancelling of systematic bias.

As reaction with OH is the predominant sink for most atmospheric trace-gases, its concentration largely defines the oxidizing power of the atmosphere (Lelieveld et al., 2004; Lelieveld et al., 2008; Lelieveld et al., 2016b) and even changes of a few

percent in its concentration are significant. An increase in the rate coefficient of the title reaction will reduce the atmospheric abundance of this centrally important radical. In Fig. S9 we illustrate the impact of using the parameterization of $k_1$ from the present study compared to the IUPAC and NASA recommendations. The upper panel in Fig. S9 plots the ratio of OH concentrations obtained when using the IUPAC parameterization and that from the present study, OH(IUPAC) / OH(this work). Throughout the troposphere OH(IUPAC) / OH(this work) deviates by only a few percent, with a value of 1.02 at the

surface and 0.96 at the tropical tropopause. OH(NASA) / OH(this work) is also 1.02 at the surface but increases to 1.04 at the tropical tropopause as the NASA-derived value of $k_1$ is lower at the temperatures and pressures encountered in this part of the atmosphere. The weak effect of changing $k_1$ on OH at the surface reflects the fact that many reactions apart from that with $NO_2$ contribute to the overall sink term for OH in the lower troposphere.

Although our experiments do not give insight into the branching between formation of HOONO and $HNO_3$ in the title reaction,

previous work predicts a significant yield of HOONO especially at low temperatures (see Fig S7). As the lifetime of HOONO with respect to re-dissociation to reactants is short et e.g. boundary layer temperatures ($\approx$ 1s at 298 K and 1 bar pressure), its formation may be seen as an effective reduction in the rate coefficient for OH + $NO_2$ (Golden and Smith, 2000). However, its lifetime increases to several days at temperatures and pressures conditions typical e.g. of the tropical tropopause (100 mbar, 220 K). As HOONO formation and loss are now parameterized (see above) in EMAC, we can explore its potential contribution

to odd-nitrogen species in the atmosphere. The reaction between OH and $NO_2$ to form HOONO converts short lived HO$x$ (HO$x$ = OH + $HO_2$) and $NO_X$ ($NO_X$ = NO + $NO_2$) into a longer lived "reservoir" species, and in this sense is similar to the reaction between $HO_2$ and $NO_2$ to form $HO_2NO_2$

$HO_2 + NO_2 + M \qquad \rightarrow HO_2NO_2 + M$                                                      (R9)

which is also thermally unstable, dissociating to reform $HO_2$ and $NO_2$. Unlike HOONO, for which there are no atmospheric

measurements, much effort has been made to measure concentrations of $HO_2NO_2$ in colder regions of the atmosphere and it is considered an important component of the $NO_y$ budget at high altitudes (Nault et al., 2016). We therefore compared EMAC predictions of HOONO concentrations with those of $HO_2NO_2$. The results are displayed in Fig. 8, in which we plot the zonally averaged HOONO / $HO_2NO_2$ ratio. Immediately apparent from Fig. 8 is the fact that compared to $HO_2NO_2$, HOONO is a minor component of $NO_y$ in the warm, lower atmosphere. This reflects the difference in the thermal decomposition rate

constant of the two trace gases, that of $HO_2NO_2$ being $\approx 4 \times 10^{-5}$ s$^{-1}$ in e.g. the middle troposphere at 400 mbar and 250 K, whereas HOONO decomposes a factor 30 faster so that its lifetime is only $\approx$ 1000 s. In the UTLS region, the ratio increases further ($HO_2NO_2$ is a factor 50 more long-lived w.r.t. thermal decomposition at 100 mbar and 220 K) but the lifetimes of both gases under these conditions are sufficiently long that their concentrations are largely determined by their production rates and their losses due to photolysis and reaction with OH. The maximum ratio of HOONO to $HO_2NO_2$ is found at the tropical

tropopause, where concentrations become comparable. As the modelled loss processes of HOONO and $HO_2NO_2$ (rate constants for photolysis and reaction with OH) are assumed to be identical, the occurrence of the maximum HOONO to $HO_2NO_2$ ratio at the tropical tropopause is related to the ratio of the (temperature dependent) rate coefficients responsible for their formation (at 220 K and 100 mbar this favours HOONO formation by a factor of $\approx$ 2) and the model OH / $HO_2$ ratio. Whilst this result indicates that HOONO could be an important reservoir of $NO_X$ under certain conditions, we must bear in mind that there is

great uncertainty associated not only with the branching ratio to HOONO formation in R1b but also with its loss processes (reaction with OH, photolysis), which remain unexplored experimentally. OH reacts with $HO_2NO_2$ via H-abstraction from the H-OO group (IUPAC, 2019), and a similar mechanism is likely for HOONO. As the H-OO bond strength is likely to be greater in HOONO than in $HO_2NO_2$ (larger electron density around the peroxy bond) we may expect the rate coefficient to be lower



for HOONO. A significantly lower rate coefficient for reaction with OH (or photolysis rate constant) could greatly increase
the abundance of HOONO. If this were the case, airborne instruments that measure $NO_x$ would likely also measure some
fraction of HOONO following its rapid decomposition in warm inlet lines, as has been observed for $HO_2NO_2$ and $CH_3O_2NO_2$
(Nault et al., 2015; Silvern et al., 2018). Clearly, more experimental or theoretical data that better constrain the yield of
HOONO and its atmospheric loss processes as well as atmospheric measurements are necessary in order to improve our
understanding of the role of the reaction between OH and $NO_2$ throughout the atmosphere.

## 4 Conclusions

We have made very precise and accurate measurements for the overall rate coefficient, $k_1$, of the reaction between OH and
$NO_2$, which is of critical importance in atmospheric chemistry. Our experiments demonstrate clearly that the presence of $H_2O$
increases significantly the overall rate coefficient ($k_1$) of the reaction between OH and $NO_2$. $H_2O$ is found to be a more efficient
collisional quencher (by a factor of $\approx 6$) of the initially formed $HO-NO_2$ association complex than $N_2$ and a factor $\approx 8$ more
efficient than $O_2$. A new parameterisation of the rate coefficient for the title reaction that considers the roles of $N_2$, $O_2$ and $H_2O$
as third-body quenchers (also using data from our companion paper, Amedro et al. (2019)) has been incorporated into a global
chemistry transport model to assess its impact on e.g. the $HNO_3$ / $NO_2$ ratio as well as $NO_X$ and OH levels. Compared to
existing evaluations of the kinetic data, use of the new parameters will result in significant changes (5-10%) in the partitioning
of $NO_X$ and $NO_Y$, the direction of the bias depending on which evaluation is used as reference and on region of the atmosphere.
This work highlights the continuing importance of obtaining accurate laboratory kinetic data for those reactions that are central
to our understanding of atmospheric chemistry and which provide anchor-points in chemical transport models.
Though the result is associated with great uncertainty owing to missing kinetic parameters for HOONO, the model predicts
the presence of HOONO in concentrations similar to those of $HO_2NO_2$ at the tropical tropopause. The present dataset addresses
only the overall rate coefficient, $k_1$. Detailed experimental studies of the formation of HOONO (e.g. its yield at various
temperatures and in the presence of $H_2O$) and on the fate of HOONO (OH kinetics, photolysis) are required to better assess its
role as $NO_X$ and HO$x$ reservoir in cold parts of the atmosphere.

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




**Table 1. Measurements of $k_1$ in He bath-gas at room temperature**


| $p$ (Torr) | M[a] | OH precursor | $k_1$[b] |
|---|---|---|---|
| 25.1 | 0.83 | $H_2O_2$ [c] | 0.75 ± 0.07 |
| 50.0 | 1.65 | $H_2O_2$ | 1.37 ± 0.08 |
| 75.1 | 2.47 | $H_2O_2$ | 1.88 ± 0.12 |
| 102.9 | 3.39 | $HNO_3$ | 2.32 ± 0.15 |
| 206.9 | 6.81 | $HNO_3$ [d] | 3.73 ± 0.25 |
| 300.7 | 9.89 | $HNO_3$ | 4.64 ± 0.29 |
| 405.8 | 13.35 | $HNO_3$ | 5.54 ± 0.37 |
| 495.6 | 16.30 | $HNO_3$ | 6.29 ± 0.40 |
| 595.0 | 19.57 | $HNO_3$ | 6.83 ± 0.42 |
| 689.1 | 22.67 | $HNO_3$ | 7.46 ± 0.46 |

[a] Units of $10^{18}$ molecule cm$^{-3}$. [b] Units of $10^{-12}$ cm$^3$ molecule$^{-1}$ s$^{-1}$. The errors are 2σ total uncertainty. [c]Concentration range of $H_2O_2$ ≈ 5-14 × $10^{13}$ molecule cm$^{-3}$. [d]Concentration range of $HNO_3$ ≈ 5-9 × $10^{13}$ molecule cm$^{-3}$.





**Table 2. Measurements of $k_1$ in $N_2$-$H_2O$ and He-$H_2O$ bath-gas**


| $T$ / K | p (Torr) | $M^a$ | $[H_2O]^b$ | $x_{He}$ or $x_{N2}$ | $x_{H2O}$ | $k_1{}^c$ |
|---|---|---|---|---|---|---|
| \multicolumn N_2-H_2O bath gas | | | | | | |
| 292 | 50.2 | 1.65 | 0 | 1 | 0 | 2.58 ± 0.16 |
| | 50.2 | 1.66 | 0.86 | 0.950 | 0.050 | 3.07 ± 0.22 |
| | 50.0 | 1.65 | 1.62 | 0.905 | 0.095 | 3.45 ± 0.26 |
| | 50.0 | 1.65 | 2.28 | 0.866 | 0.134 | 3.83 ± 0.26 |
| | 50.2 | 1.66 | 2.84 | 0.834 | 0.166 | 3.95 ± 0.37 |
| | 49.2 | 1.63 | 3.27 | 0.805 | 0.195 | 4.10 ± 0.27 |
| | 50.0 | 1.65 | 4.06 | 0.754 | 0.246 | 4.47 ± 0.18 $^d$ |
| \multicolumn He-H_2O bath gas | | | | | | |
| 277 | 48.6 | 1.68 | 0 | 1 | 0 | 1.59 ± 0.11 |
| | 47.6 | 1.66 | 0.9 | 0.946 | 0.054 | 2.27 ± 0.15 |
| | 48.0 | 1.67 | 1.42 | 0.915 | 0.085 | 2.63 ± 0.17 |
| | 48.7 | 1.7 | 2 | 0.882 | 0.118 | 3.13 ± 0.24 |
| 291 | 50.0 | 1.65 | 0 | 1 | 0 | 1.37 ± 0.08 |
| | 50.6 | 1.68 | 0.64 | 0.962 | 0.038 | 1.99 ± 0.14 |
| | 51 | 1.69 | 1.30 | 0.923 | 0.077 | 2.39 ± 0.21 |
| | 50.7 | 1.68 | 2.25 | 0.863 | 0.137 | 2.88 ± 0.24 |
| | 49.5 | 1.64 | 3.06 | 0.818 | 0.182 | 3.43 ± 0.22 |
| | 50.8 | 1.68 | 3.12 | 0.810 | 0.190 | 3.44 ± 0.24 |
| | 49.7 | 1.65 | 3.60 | 0.783 | 0.217 | 3.54 ± 0.23 |
| | 50.2 | 1.66 | 3.94 | 0.764 | 0.236 | 3.72 ± 0.29 |
| | 50.5 | 1.67 | 4.68 | 0.721 | 0.279 | 4.08 ± 0.27 |
| 332 | 56.8 | 1.65 | 0 | 1 | 0 | 0.99 ± 0.06 |
| | 56.3 | 1.64 | 0.58 | 0.964 | 0.036 | 1.32 ± 0.08 |
| | 56 | 1.63 | 1.72 | 0.895 | 0.105 | 1.81 ± 0.16 |
| | 56.2 | 1.63 | 3.3 | 0.798 | 0.202 | 2.43 ± 0.18 |
| | 55.9 | 1.62 | 4.33 | 0.733 | 0.267 | 2.88 ± 0.22 |

Unless otherwise indicated, the measurements were performed using $H_2O_2$ as OH precursor. The concentration range of $H_2O_2$ was 5-18 × $10^{13}$ molecule cm$^{-3}$ for experiments in He-$H_2O$ bath gas and 9-14 × $10^{13}$ molecule cm$^{-3}$ for experiments in He-$H_2O$ bath gas. $^a$Units of $10^{18}$ molecule cm$^{-3}$. $^b$Units of $10^{17}$ molecule cm$^{-3}$. $^c$Units of $10^{-12}$ cm$^3$ molecule$^{-1}$ s$^{-1}$. Errors are 2σ

total uncertainty. $^d$ measurement performed using $O_3$-$H_2O$ as OH precursor (with $[O_3]$ = 2 × $10^{13}$ molecule cm$^{-3}$).





**Table 3. Parameters for calculating $k_1$ using Eqn. (15) and (16)**

| Bath-gas | $k_0{}^a$ | $T$-dependence of $k_0$ ($m$, $q$ or $o$) | $k_\infty{}^b$ | $F_c$ |
|---|---|---|---|---|
| $N_2$ | $2.6 \times 10^{-30}$ | 3.6 ($m$) | | |
| $O_2$ | $2.0 \times 10^{-30}$ | 3.6 ($q$) | $6.3 \times 10^{-11}$ | 0.39 |
| $H_2O$ | $15.9 \times 10^{-30}$ | 3.4 ($o$) | | |

[a] Units of $cm^6$ molecule$^{-2}$ s$^{-1}$. [b] Units of $cm^3$ molecule$^{-1}$ s$^{-1}$. Note that $k_\infty$ is independent of temperature ($n = 0$).





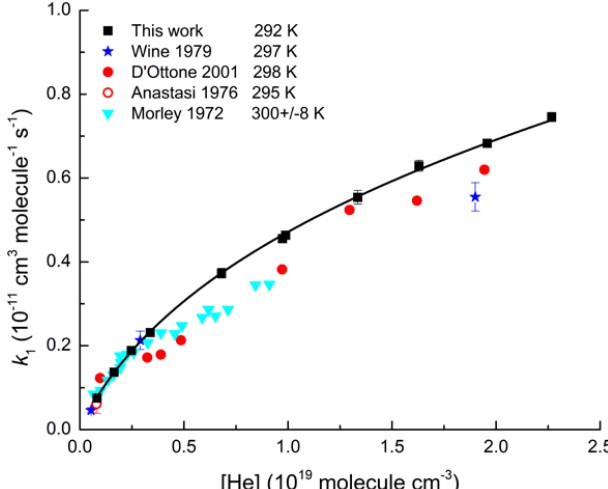

**Figure 1.** Values of $k_1$ from this study (black squares) as a function of He concentration at 292 K. Errors are 2σ statistical only. The solid line is a fit to our data using Eqn. (4) with $k_0 = 1.4 \times 10^{-30}$ cm$^6$ molecule$^{-2}$ s$^{-1}$, $k_\infty = 6.3 \times 10^{-11}$ cm$^3$ molecule$^{-1}$ s$^{-1}$ and $F_c = 0.32$. Previous datasets at room temperature (Wine et al. (1979), D'Ottone et al. (2001), Anastasi and Smith (1976) and Morley and Smith (1972)) are displayed for comparison.



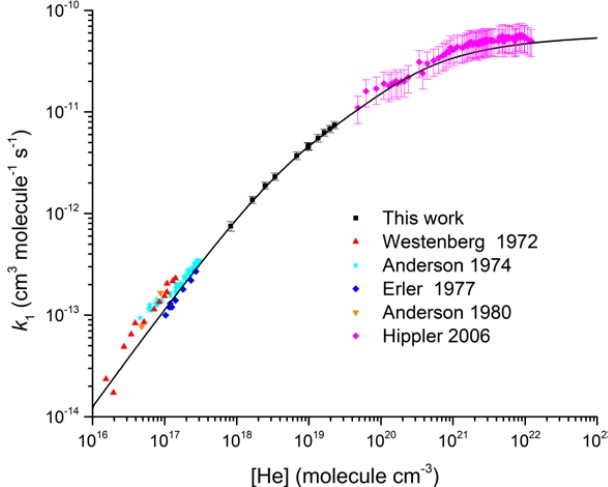

**Figure 2**. Comparison between the present dataset, the high pressure measurements by Hippler et al. (2006) and the low pressure measurements by Anderson et al. (1974), Westenberg and Dehaas (1972), Anderson (1980) and Erler et al. (1977). All measurements were made at room-temperature. The black line is our parameterisation with $k_0 = 1.4 \times 10^{-30}$ cm$^6$ molecule$^{-2}$ s$^{-1}$, $k_\infty = 6.3 \times 10^{-11}$ cm$^3$ molecule$^{-1}$ s$^{-1}$, $m = 3.1$ and $F_c = 0.32$.






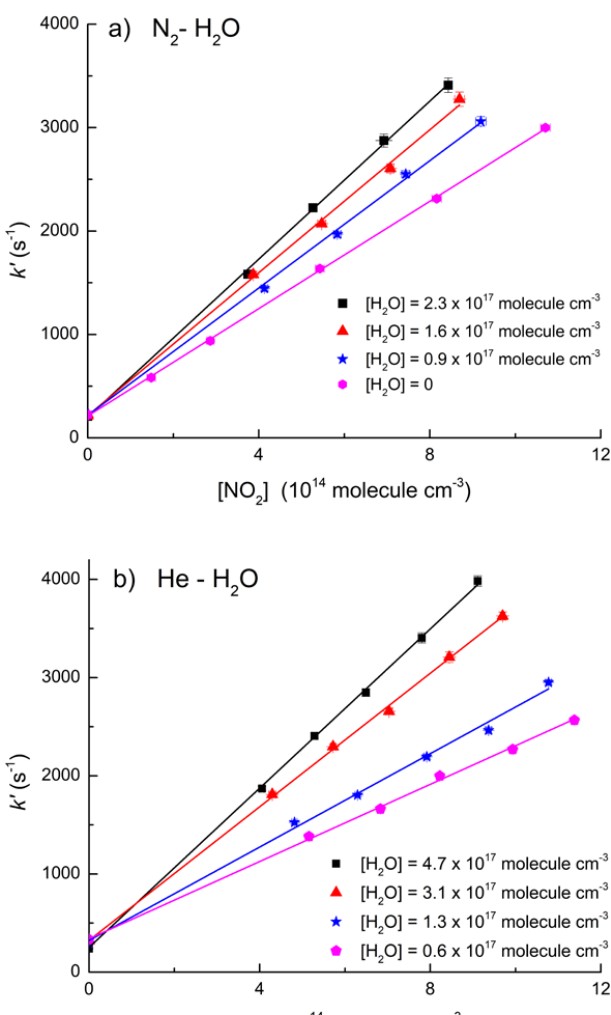

**Figure 3 a)** Data obtained in $N_2$-$H_2O$ bath-gas (50 Torr, 292 K). **b)** Data obtained in He-$H_2O$ bath-gas (50 Torr, 291 K). Both panels display first-order, OH decay constants in various concentrations of $NO_2$ and different mole fractions of $H_2O$. The solid lines represent least squares linear fits to Eqn. (2).

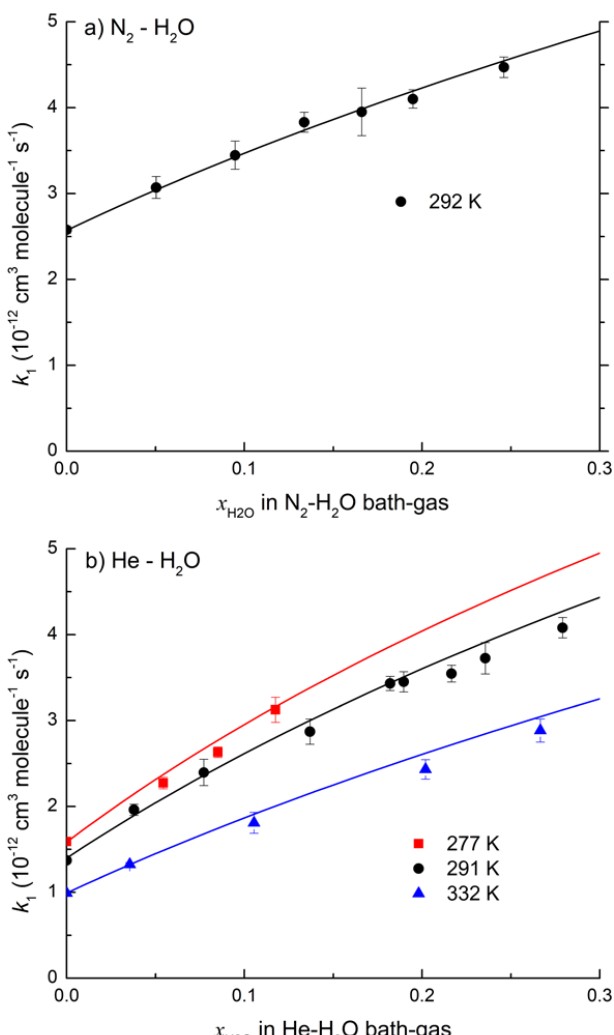

**Figure 4. a)** $k_1$ as a function of $x_{H2O}$ at 50 Torr $N_2$-$H_2O$ and 292 K. The line represents a least squares, multivariate fit (Eqn. 7 and 8) with $k_\infty = 6.3 \times 10^{-11}$ cm$^3$ molecule$^{-2}$ s$^{-1}$, $k_0^{N2} = 2.6 \times 10^{-30}$ cm$^6$ molecule$^{-2}$ s$^{-1}$, $F_c = 0.39$, $m = 3.6$, $k_0^{H2O} = 15.9 \times 10^{-30}$ cm$^6$ molecule$^{-2}$ s$^{-1}$, $o = 3.4$. **b)** $k_1$ as a function of $x_{H2O}$ in He-$H_2O$ mixtures at 277, 291 and 332 K. The solid lines represent a least squares, multivariate fit (Eqn. 7 and 9 to 12) where $k_\infty = 6.3 \times 10^{-11}$ cm$^3$ molecule$^{-2}$ s$^{-1}$, $k_0^{He} = 1.4 \times 10^{-30}$ cm$^6$ molecule$^{-2}$ s$^{-1}$, $F_c^{He} = 0.32$, $m = 3.1$, $k_0^{H2O} = 15.9 \times 10^{-30}$ cm$^6$ molecule$^{-2}$ s$^{-1}$, $F_c^{H2O} = 0.39$ and $o = 3.4$.




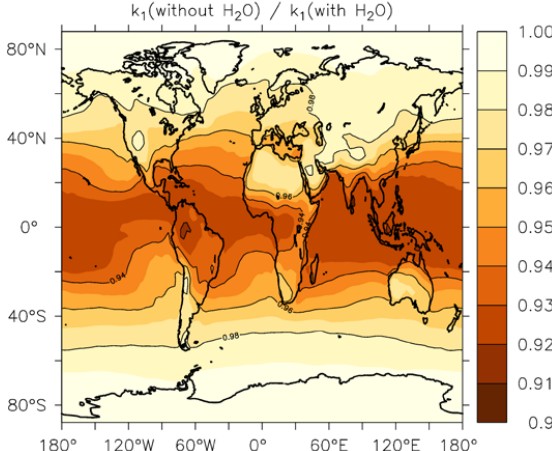

**Figure 5.** Annual average effect of $H_2O$ on $k_1$ expressed as the fractional change in the rate coefficient at the Earth's surface when setting the mole fraction of water vapour to zero in Eqn. 15.



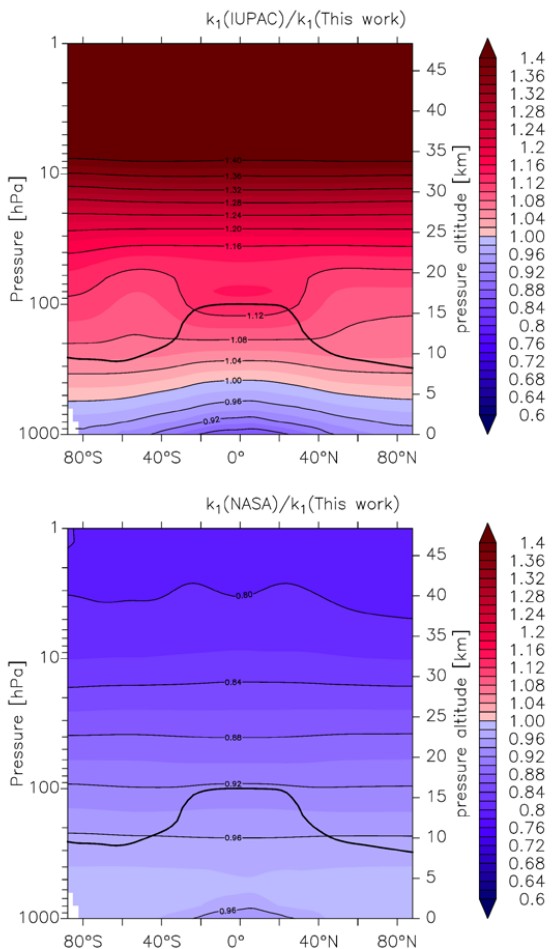

**Fig. 6** Global values of $\frac{k_1^{\text{IUPAC}}}{k_1^{\text{this work}}}$ (upper panel) and $\frac{k_1^{\text{NASA}}}{k_1^{\text{this work}}}$ lower panel). $k_1$ is the overall rate coefficient (both channels)

for Reaction R1 calculated using the parameters from this work ($k_1^{\text{this work}}$) and those presently recommended by the IUPAC

($k_1^{\text{IUPAC}}$) and NASA ($k_1^{\text{NASA}}$) data evaluation panels. The black line represents the model tropopause.






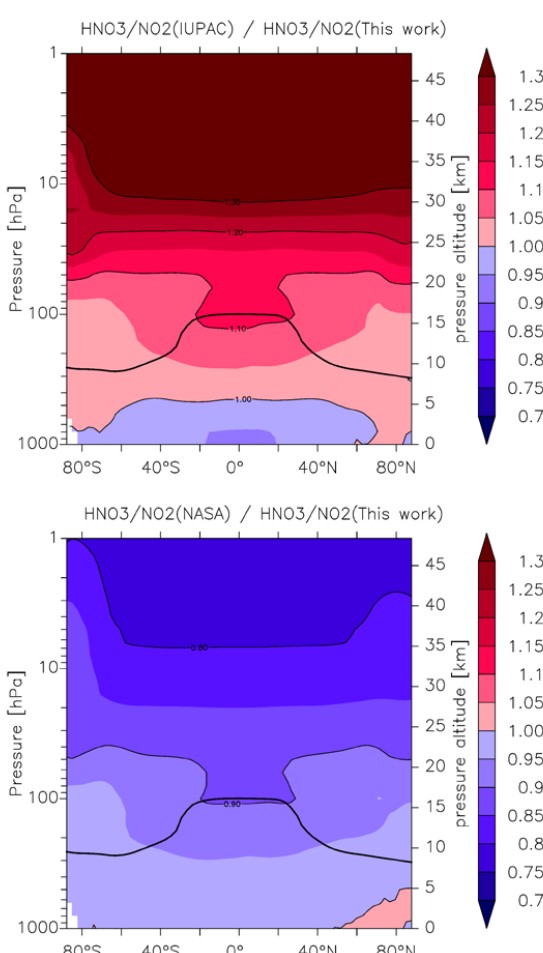

**Figure 7.** Effect of different parameterisations of $k_1$ on the global (zonal and yearly averaged) $HNO_3$ to $NO_2$ ratio. The upper panel plots $\frac{HNO3}{NO2}$ (IUPAC)$/\frac{HNO3}{NO2}$ (this work), the lower panel plots $\frac{HNO3}{NO2}$ (NASA)$/\frac{HNO3}{NO2}$ (this work). The black line represents the model tropopause.



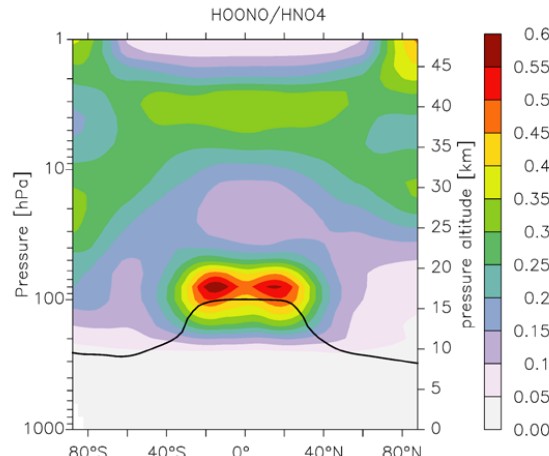

**Figure 8.** Model (EMAC) ratio of HOONO (formed in the reaction of $NO_2$ with OH) to $HO_2NO_2$ (formed in the reaction of $NO_2$ with $HO_2$) calculated using the present parameterisation of $k_1$ and equating the (unknown) rate coefficients for loss of HOONO via reaction with OH or photolysis to those of $HO_2NO_2$. The black line represents the model tropopause.
