# Peer review of "Kinetics of the OH + NO2 reaction: Effect of water vapour and new parameterisation for global modelling."

_Atmospheric Chemistry and Physics, 2019_

## Referee Comment (RC1) · Anonymous Referee #1 · 19 Dec 2019

This is an excellent and very careful study of the kinetics of the reaction of OH radical with NO2, a reaction of central atmospheric significance. The main finding associated with this work (the second paper from this group on this reaction) is the finding of a significant enhancement to the OH/NO2 rate coefficient in the presence of water vapor. This enhancement is quantified in both N2 and He bath gas, and as a function of temperature. The data appear to be of the highest quality, previous data are discussed in detail, and the modeling work adds significantly to the atmospheric context, and hence the overall value of the manuscript. Overall a superb study! I recommend publication in ACP, after consideration of the minor comments listed below.

[Figure]

L12 – molecule misspelled.

L64 – exponent should be +3.

L127 – evaluation of atmospheric

L148- Maybe it is obvious, but it could be added to the text that fitting with an increased $F_c$ gives a lower $k_o$, (which doesn't fit the data).

Page 4,5 - The pure He data are taken here at 292 K, somewhat below other 'room temperature' datasets that comparisons are being made with. Has this has been taken into account? If not, could this account for at least some of the systematic discrepancies with D'Ottoni or Morley? On a related point, there is a T-dependence to the He $k_o$ that appears in various places (captions to Fig 2, Fig 4b, L252). What is the origin of this value? (I think it is mostly unnecessary).

L209- delete molecules

L252 – temperature dependence

L254 – delete use

L275 and forward – The tests for HONO and HNO3 are really nice !

L297 – temperature instead of temperatures

L342 – H2O, not HO2, as product of (R7)

L406 – short at

L408 – temperature and pressure conditions

---

## Referee Comment (RC2) · Anonymous Referee #2 · 23 Dec 2019

The manuscript describes the determination of the rate constant of the OH + NO2 reaction with He and N2 as bath gases in presence and absence of gaseous H2O. A quasi-static reaction cell was used, and OH was produced by pulsed laser photolysis of HNO3, H2O2, or O3/H2O mixtures. Pseudo-first order conditions with respect to [OH] were applied. The OH concentration was monitored time-resolved by laser-induced fluorescence, and the (crucial) NO2 concentration was carefully determined with two different absorption-spectroscopic approaches. A notable increase of the OH + NO2 rate constant in He and N2 when H2O is present was observed and associated with a particularly high efficiency of H2O for collisional stabilization of the HNO3 product. Non-linear mixing rules for the collisional efficiencies seem to apply. Very careful

parameterizations and statistical evaluations of the experimental results, including earlier literature data, were performed and discussed in great detail, also with respect to branching between HONO2 and HOONO as reaction products. The newly parameterized rate constant is incorporated in a 3D chemical transport model, and effects on quantities such as the atmospheric HNO3/NO2 ratio, the atmospheric concentration of OH, or the HOONO/HO2NO2 ratio are assessed. All in all, this is a very nice paper bridging high-level state-of-the-art laboratory measurements with global atmospheric modeling. So the topic is at the very heart of ACP, and I recommend publication essentially 'as is' with only very few, very minor points to be considered by the authors:

line 35: 'gases' should probably read 'gas'

line 68: 'O3-H2O ' should probably better read 'O3/H2O'

Tables 1 and 2: please specify/explain M

Fig. 1, figure caption: please give the parameters m and n

Fig. 2, figure caption: please give the parameter n

---

## Referee Comment (RC3) · Anonymous Referee #3 · 29 Dec 2019

See attached

Please also note the supplement to this comment:
https://www.atmos-chem-phys-discuss.net/acp-2019-1103/acp-2019-1103-RC3-supplement.pdf

———————————————————

[Figure]

Review of Amedro et al

This manuscript presents the first study of the effectiveness of $H_2O$ as a third body on the recombination of OH with $NO_2$. This careful experimental study shows that water vapor is much more effective than $N_2$ or $O_2$ in causing recombination. It also presents a global modeling study of a new parameterization of the OH + $NO_2$ reaction as compared to the IUPAC and JPL recommendations; this parameterization uses the results of a previous study showing that $O_2$ and $N_2$ have different efficiencies in quenching the products of OH + $NO_2$. The modeling suggests that HOONO could be a non-negligible reservoir of NOx in some parts of the atmosphere. This is a very important paper that is clearly in the scope of ACP. There are no major problems with the manuscript, but a few of points should be clarification or emphasized more strongly before publication in ACP.

My major concern about this manuscript is actually rather minor: In the global modeling, it is not clear how much of the affects of the new parameterization, occurs due to water vapor and how much due to the use of the results of the author's previous paper on $N_2$ vs $O_2$ as colliders. This should be made clear.

The enhancement of the quenching of the energized $HNO_3$ intermediate ($HNO_3^*$) due to $H_2O$ vapor is presumably due to the strong hydrogen bonding between the two (stronger than OH-$H_2O$ or $NO_2$-$H_2O$). It would be good to make this explicit and add some references to the literature on the $HONO_2$-HOH complex.

There must be previous field work measuring [NO2]/[$HONO_2$] and corresponding modeling work that did or did not find discrepancies. It seems that the authors should refer discuss a few of these, at least briefly.

Minor Issues:

Line 12: "molecule" is missing an "l"

Line 45: "being" should be "is"

Line 50-52. The sentence beginning "Theoretical calculations…" might better appear immediately after the discussion of the chaperone mechanism, rather than after the introduction of enhanced collider gases.

line 68 "prevented" should be "preventing"

line 75: "in Tables 1 and 2." might better be phrased as "in the notes to Tables 1 and 2."

It might help orient readers if the manuscript provided some idea of the conditions under which the OH + NO2 → HONO2 is nearly in the low-pressure limit and high-pressure limit.

Line 102: The manuscript states that the low vapor pressure of water prevents it from being used as a bath gas by itself, but 5 Torr of water vapor is roughly equivalent of 50 Torr of He. So I

**Fig. 1.**

**Supplement:**

Review of Amedro et al

This manuscript presents the first study of the effectiveness of $H_2O$ as a third body on the recombination of OH with $NO_2$. This careful experimental study shows that water vapor is much more effective than $N_2$ or $O_2$ in causing recombination. It also presents a global modeling study of a new parameterization of the OH + $NO_2$ reaction as compared to the IUPAC and JPL recommendations; this parameterization uses the results of a previous study showing that $O_2$ and $N_2$ have different efficiencies in quenching the products of OH + $NO_2$. The modeling suggests that HOONO could be a non-negligible reservoir of NOx in some parts of the atmosphere. This is a very important paper that is clearly in the scope of ACP. There are no major problems with the manuscript, but a few of points should be clarification or emphasized more strongly before publication in ACP.

My major concern about this manuscript is actually rather minor: In the global modeling, it is not clear how much of the affects of the new parameterization, occurs due to water vapor and how much due to the use of the results of the author's previous paper on $N_2$ vs $O_2$ as colliders. This should be made clear.

The enhancement of the quenching of the energized $HNO_3$ intermediate ($HNO_3$*) due to $H_2O$ vapor is presumably due to the strong hydrogen bonding between the two (stronger than OH-$H_2O$ or $NO_2$-$H_2O$). It would be good to make this explicit and add some references to the literature on the $HONO_2$-HOH complex.

There must be previous field work measuring [NO2]/[$HONO_2$] and corresponding modeling work that did or did not find discrepancies. It seems that the authors should refer discuss a few of these, at least briefly.

Minor Issues:

Line 12: "molecule" is missing an "l"

Line 45: "being" should be "is"

Line 50-52. The sentence beginning "Theoretical calculations…" might better appear immediately after the discussion of the chaperone mechanism, rather than after the introduction of enhanced collider gases.

line 68 "prevented" should be "preventing"

line 75: "in Tables 1 and 2." might better be phrased as "in the notes to Tables 1 and 2."

It might help orient readers if the manuscript provided some idea of the conditions under which the OH + NO2 $\rightarrow$ $HONO_2$ is nearly in the low-pressure limit and high-pressure limit.

Line 102: The manuscript states that the low vapor pressure of water prevents it from being used as a bath gas by itself, but 5 Torr of water vapor is roughly equivalent of 50 Torr of He. So I

think that it would be clearer to say that it is not possible to determine $k_0(H2O)$ by using pure water vapor as a bath gas.

Lines 205 ff. "In other words…." It is not clear to this reader how it follows from the previous text that the total rate constant of a $H_2O$-$N_2$ bath gas is not the sum of individual rate constants $k(P_i,T)$, where i=$H_2O$ or $N_2$. It seems like a step of the logic has not been made explicit, and that it would help the reader if the manuscript made the logic clearer.

In Section 3.2.1, it would be helpful to indicate the pressures at which $k(P,300)$ deviates by more than 10% from the low-pressure and high-pressure limits. This would help orient the reader.

Line 298: the pressures only add up to 990 mbar, not 1 bar.

Lines 334 ff. If I understand correctly, the manuscript takes the branching ratio between reactions (1a) and (1b) from previous work. This is equivalent to assuming that water vapor enhances both rate constants to the same extent. This assumption should be made very explicit in the manuscript.

Lines 432-433: Are the H-OONO and H-OONO2 bond energies known from computational chemistry (well enough to determine which is stronger)?

Figure 3 lacks error bars.

Table 2: The caption lists the range of [HOOH] used for the He-$H_2O$ experiments twice. I suspect one of these is for the $N_2$-$H_2O$ experiments.

---

## Author Comment (AC1) · 17 Jan 2020

The following contains the comments of the referee (black), our replies (blue) indicating changes that will be made to the revised document (red).

**Reviewer #1**

This is an excellent and very careful study of the kinetics of the reaction of OH radical with NO2, a reaction of central atmospheric significance. The main finding associated with this work (the second paper from this group on this reaction) is the finding of a significant enhancement to the OH/NO2 rate coefficient in the presence of water vapor. This enhancement is quantified in both N2 and He bath gas, and as a function of temperature. The data appear to be of the highest quality, previous data are discussed in detail, and the modeling work adds significantly to the atmospheric context, and hence the overall value of the manuscript. Overall a superb study! I recommend publication in ACP, after consideration of the minor comments listed below.
We thank the reviewer for the careful review and the positive assessment of our manuscript.

L12 – molecule misspelled. Corrected

L64 – exponent should be +3. Corrected

L127 – evaluation of atmospheric. Corrected

L148- Maybe it is obvious, but it could be added to the text that fitting with an increased Fc gives a lower ko, (which doesn't fit the data).
We have added:
We note that using a higher $F_c$=0.39 resulted in a lower value of $k_0^{He}$ equal to $1.0 \times 10^{-30}$ cm$^6$ molecule$^{-2}$ s$^{-1}$.

Page 4,5 - The pure He data are taken here at 292 K, somewhat below other 'room temperature' datasets that comparisons are being made with. Has this has been taken into account? If not, could this account for at least some of the systematic discrepancies with D'Ottoni or Morley?
In Figure 1, we did not attempt to apply any corrections to previous works to account for temperature differences. In the more rigorous comparison of our work with others (Fig S3-S5), we used our parametrization at the given temperatures of the previous studies.
We now write in the Figure S3, S4 and S5 captions the temperature for which $k(T,p)$ was calculated.

On a related point, there is a T-dependence to the He ko that appears in various places (captions to Fig 2, Fig 4b, L252). What is the origin of this value? (I think it is mostly unnecessary).
The T-dependence in He was determined in our laboratory over a small range of temperatures (277, 292 and 332 K) and pressures. We have added the data points used to determine $m$(He) in Table 1, a Figure (S10) in the supplementary information and the text below:

The T-dependence factor in He, $m$(He), was determined to be 3.1 over the temperature range from 277 to 332 K (Table 1 and Figure S6).

| $T$ (K) | $p$ (Torr) | M[a] | OH precursor | $k_1$[b] |
|---|---|---|---|---|
| 277 | 48.6 | 1.68 | $H_2O_2$ | $1.59 \pm 0.14$ |
| 292 | 25.1 | 0.83 | $H_2O_2$ [c] | $0.75 \pm 0.07$ |
| | 50.0 | 1.65 | $H_2O_2$ | $1.37 \pm 0.08$ |
| | 75.1 | 2.47 | $H_2O_2$ | $1.88 \pm 0.12$ |
| | 102.9 | 3.39 | $HNO_3$ | $2.32 \pm 0.15$ |
| | 206.9 | 6.81 | $HNO_3$ [d] | $3.73 \pm 0.25$ |
| | 300.7 | 9.89 | $HNO_3$ | $4.64 \pm 0.29$ |
| | 405.8 | 13.35 | $HNO_3$ | $5.54 \pm 0.37$ |
| | 495.6 | 16.30 | $HNO_3$ | $6.29 \pm 0.40$ |
| | 595.0 | 19.57 | $HNO_3$ | $6.83 \pm 0.42$ |
| | 689.1 | 22.67 | $HNO_3$ | $7.46 \pm 0.46$ |
| 332 | 28.1 | 0.82 | $H_2O_2$ | $0.60 \pm 0.06$ |
| | 56.8 | 1.65 | $H_2O_2$ | $0.99 \pm 0.08$ |
| | 85.4 | 2.48 | $H_2O_2$ | $1.34 \pm 0.10$ |

[Figure]

Figure S6. Values of $k_1$ as a function of He concentration at 277, 292 and 332 K. Errors are 2σ statistical only. The solid line is a fit to our data using Eqn. (4) with $k_0 = 1.4 \times 10^{-30}$ cm$^6$ molecule$^{-2}$ s$^{-1}$, $k_\infty = 6.3 \times 10^{-11}$ cm$^3$ molecule$^{-1}$ s$^{-1}$, $F_c = 0.32$, $m = 3.1$ and $n = 0$.

L209- delete molecules Corrected

L252 – temperature dependence Corrected

L254 – delete use Corrected

L275 and forward – The tests for HONO and HNO3 are really nice !

L297 – temperature instead of temperatures Corrected

L342 – H2O, not HO2, as product of (R7) Corrected

L406 – short at Corrected

L408 – temperature and pressure conditions Corrected

---

## Author Comment (AC2) · 17 Jan 2020

The following contains the comments of the referee (black), our replies (blue) indicating changes that will be made to the revised document (red).

**Reviewer #2**

The manuscript describes the determination of the rate constant of the OH + NO2 reaction with He and N2 as bath gases in presence and absence of gaseous H2O. A quasi-static reaction cell was used, and OH was produced by pulsed laser photolysis of HNO3, H2O2, or O3/H2O mixtures. Pseudo-first order conditions with respect to [OH] were applied. The OH concentration was monitored time-resolved by laser-induced fluorescence, and the (crucial) NO2 concentration was carefully determined with two different absorption-spectroscopic approaches. A notable increase of the OH + NO2 rate constant in He and N2 when H2O is present was observed and associated with a particularly high efficiency of H2O for collisional stabilization of the HNO3 product. Non-linear mixing rules for the collisional efficiencies seem to apply. Very careful parameterizations and statistical evaluations of the experimental results, including earlier literature data, were performed and discussed in great detail, also with respect to branching between HONO2 and HOONO as reaction products. The newly parameterized rate constant is incorporated in a 3D chemical transport model, and effects on quantities such as the atmospheric HNO3/NO2 ratio, the atmospheric concentration of OH, or the HOONO/HO2NO2 ratio are assessed. All in all, this is a very nice paper bridging high-level state-of-the-art laboratory measurements with global atmospheric modeling. So the topic is at the very heart of ACP, and I recommend publication essentially 'as is' with only very few, very minor points to be considered by the authors:
We thank the reviewer for the careful review and the positive assessment of our manuscript.

line 35:'gases' should probably read 'gas' Corrected

line 68:'O3-H2O' should probably better read 'O3/H2O' Corrected

Tables 1 and 2: please specify/explain M
In Table 1, we modified the caption below with:
Molecular density M(He) in units of $10^{18}$ molecule cm$^{-3}$
In Table 2, we modified in the caption,
Molecular density M(He-H$_2$O) or M(N$_2$-H$_2$O) in units of $10^{18}$ molecule cm$^{-3}$

Fig. 1, figure caption: please give the parameters m and n
The figure 1 caption now reads:
The solid line is a fit to our data using Eqn. (4) with $k_0 = 1.4 \times 10^{-30}$ cm$^6$ molecule$^{-2}$ s$^{-1}$, $k_\infty = 6.3 \times 10^{-11}$ cm$^3$ molecule$^{-1}$ s$^{-1}$, $F_c = 0.32$, $m=3.1$ and $n=0$.

Fig. 2, figure caption: please give the parameter n
The figure 2 caption now reads:
The black line is our parameterisation with $k_0 = 1.4 \times 10^{-30}$ cm$^6$ molecule$^{-2}$ s$^{-1}$, $k_\infty = 6.3 \times 10^{-11}$ cm$^3$ molecule$^{-1}$ s$^{-1}$, $m = 3.1$, $n=0$ and $F_c = 0.32$.

---

## Author Comment (AC3) · 17 Jan 2020

The following contains the comments of the referee (black), our replies (blue) indicating changes that will be made to the revised document (red).

**Reviewer #3**

This manuscript presents the first study of the effectiveness of H2O as a third body on the recombination of OH with NO2. This careful experimental study shows that water vapor is much more effective than N2 or O2 in causing recombination. It also presents a global modeling study of a new parameterization of the OH + NO2 reaction as compared to the IUPAC and JPL recommendations; this parameterization uses the results of a previous study showing that O2 and N2 have different efficiencies in quenching the products of OH + NO2 . The modeling suggests that HOONO could be a non-negligible reservoir of NOx in some parts of the atmosphere. This is a very important paper that is clearly in the scope of ACP. There are no major problems with the manuscript, but a few of points should be clarification or emphasized more strongly before publication in ACP.
We thank the reviewer for the careful review and the positive assessment of our manuscript.

My major concern about this manuscript is actually rather minor: In the global modeling, it is not clear how much of the affects of the new parameterization , occurs due to water vapor and how much due to the use of the results of the author's previous paper on N2 vs O2 as colliders. This should be made clear.
As we already mention, the impact of $H_2O$ is limited to the boundary-layer (Fig. 5), above which the $H_2O$ concentration decreases rapidly. We now emphasise this by writing:
The presence of water vapour does not impact on values of $k_1$ above the boundary layer.

The enhancement of the quenching of the energized HNO3 intermediate (HNO3 *) due to H2O vapor is presumably due to the strong hydrogen bonding between the two (stronger than OH-H2O or NO2-H2O). It would be good to make this explicit and add some references to the literature on the HONO2-HOH complex.
We have added text mentioning the $HNO_3 – H_2O$ complex:
Water vapour is therefore a factor $\approx 8$ more efficient than $O_2$, and a factor ~6 more efficient than $N_2$ as a quencher of the $HO-NO_2$ intermediate, which is qualitatively consistent with known strong binding (40 kJ mol$^{-1}$) in the $HNO_3–H_2O$ complex (Tao et al., 1996).

There must be previous field work measuring [NO2]/[HONO2] and corresponding modeling work that did or did not find discrepancies. It seems that the authors should refer discuss a few of these, at least briefly.
This is an old problem in atmospheric chemistry. $HNO_3$ is not formed solely in the reaction between OH and $NO_2$ but also in heterogeneous processes that hydrolyse $N_2O_5$, the rate of which depends on poorly constrained factors such as e.g. the aerosol surface area. Uncertainties in modelled OH are large as are uncertainties for $HNO_3$ measurements, which in the boundary layer reflect deposition to surfaces, again poorly constrained. In short, uncertainties in kinetic parameters are only one factor that influence $NO_2$ and $HNO_3$ ratios and this issue is too complex to deal with properly in this manuscript.

Minor Issues:

Line 12: "molecule" is missing an "l" Corrected

Line 45: "being" should be "is" Corrected

Line 50-52. The sentence beginning "Theoretical calculations..." might better appear immediately after the discussion of the chaperone mechanism, rather than after the introduction of enhanced collider gases.
Corrected as suggested

line 68 "prevented" should be "preventing" Corrected

line 75: "in Tables 1 and 2." might better be phrased as "in the notes to Tables 1 and 2."
Corrected as suggested

It might help orient readers if the manuscript provided some idea of the conditions under which the OH + NO2 →HONO2 is nearly in the low-pressure limit and high-pressure limit.
We added the following text
Under the conditions of $T$ and $p$ relevant for atmospheric chemistry, the title reaction is in the fall-off regime.

Line 102: The manuscript states that the low vapor pressure of water prevents it from being used as a bath gas by itself, but 5 Torr of water vapor is roughly equivalent of 50 Torr of He. So I think that it would be clearer to say that it is not possible to determine k0 (H2O) by using pure water vapor as a bath gas.
This is correct. We prefer to remove this statement completely.

Lines 205 ff. "In other words...." It is not clear to this reader how it follows from the previous text that the total rate constant of a H2O-N2 bath gas is not the sum of individual rate constants k(Pi,T), where i=H2O or N2. It seems like a step of the logic has not been made explicit, and that it would help the reader if the manuscript made the logic clearer.
Starting the sentence with "In other words" was indeed confusing.
We have removed this.

In Section 3.2.1, it would be helpful to indicate the pressures at which k(P,300) deviates by more than 10% from the low-pressure and high-pressure limits. This would help orient the reader.
It is not obvious what insight this brings as there is no physical meaning associated with a 10% deviation. We believe this would be confusing and prefer not to follow this suggestion.

Line 298: the pressures only add up to 990 mbar, not 1 bar.
We now write:
The pressure of water vapour is 34 mbar, those of $O_2$ and $N_2$ are then 210 and 756 mbar, respectively

Lines 334 ff. If I understand correctly, the manuscript takes the branching ratio between reactions (1a) and (1b) from previous work. This is equivalent to assuming that water vapor

enhances both rate constants to the same extent. This assumption should be made very explicit in the manuscript.

We have added the following sentence to make this clear.

We assume $\alpha$ is independent of water vapour.

Lines 432-433: Are the H-OONO and H-OONO2 bond energies known from computational chemistry (well enough to determine which is stronger)?

We are not aware of such calculations and are not in a position to do them ourselves. The text simply introduces the possibility that the OH rate coefficients for H-OONO and H-OONO2 are different and suggests that theoretical calculations would be useful. Hopefully, someone will conduct such calculations and calculate the relative rate coefficient.

Figure 3 lacks error bars.

The error bars are already on the plot but are in most case smaller than the symbol. We have added to the caption:

With a few exceptions, the error bars (2 $\sigma$) are generally smaller than the symbols.

Table 2: The caption lists the range of [HOOH] used for the He-H2O experiments twice. I suspect one of these is for the N2-H2O experiments. Correction made.